# Epibiotic bacterial community composition varies during different developmental stages of *Octopus mimus*: Study of cultivable representatives and their secondary metabolite production

**Martha B. Hengst**[1¤]*, **Stephanie Trench**[1], **Valezka Alcayaga**[2], **Cristian Sepúlveda-Muñoz**[3], **Jorge Bórquez**[4], **Mario Simirgiotis**[5], **Fernando Valenzuela**[3], **Mario Lody**[3], **Lenka Kurte**[1], **Coral Pardo-Esté**[1]

1 Laboratorio de Ecología Molecular y Microbiología Aplicada, Departamento de Ciencias Farmacéuticas, Universidad Católica del Norte, Antofagasta, Chile, 2 Universidad de Antofagasta, Antofagasta, Chile, 3 Laboratorio de Ecología Microbiana, FAREMAR, Centro de Bioinnovación, Universidad de Antofagasta, Antofagasta, Chile, 4 Laboratorio de Productos Naturales, Facultad de Ciencias Básicas, Universidad de Antofagasta, Antofagasta, Chile, 5 Institute of Pharmacy, Universidad Austral de Chile, Valdivia, Chile

¤ Current address: Departamento de Ciencias Farmacéuticas, Universidad Católica del Norte, Antofagasta, Chile
* mhengst@ucn.cl

## Abstract

Marine microbial communities colonizing the skin of invertebrates constitute the primary barrier between host and environment, potentially exerting beneficial, neutral, or detrimental effects on host fitness. To evaluate the potential contribution of epibiotic bacteria to the survival of early developmental stages of *Octopus mimus*, bacterial isolates were obtained from eggs, paralarvae, and adults. Their enzymatic activities were determined, and antibacterial properties were assessed against common marine pathogens. The isolates belonged to the phyla Proteobacteria, Actinomycetota, Bacteroidota, and Bacillota, represented by 21 genera and 27 species. Specific taxa were associated with each developmental stage, with only three species shared among different stages: *Bacillus pumilus*, *B. megaterium*, and *Shewanella algae*, which also inhibited the growth of all assayed pathogens. Organic extracts from *Bacillus megaterium* M8-1 were obtained, and UHPLC-MS analysis detected seventeen putative compounds, including two phenolic acids, three indole derivatives, and twelve oxylipins. Our findings provide novel data on cultivable bacterial representatives isolated from *Octopus mimus* capable of synthesizing chemical compounds with bioactive properties. These results contribute to a better understanding of the role of microbial communities in the survival of this invertebrate species during critical early life stages.

**Data Availability Statement:** All sequences are available from the GenBank database (under accession numbers KX218274.1, KX218275.1, KX218317, KX218318, KX218282 and AY034144.1) All other relevant data are within the paper.

**Funding:** CODEI 5388 Project UA (MB Hengst).

**Competing interests:** The authors have declared that no competing interests exist.

## 1. Introduction

Cephalopods are marine invertebrates with great potential in aquaculture due to their captivity adaptation, feeding habits, high conversion rates, reproductive characteristics, accelerated growth, and commercial profitability [1–3]. *Octopus mimus* (Cephalopoda, Octopodidade) is one of the main benthic resources supporting the artisanal fisheries in northern Chile, from which close to 90% of captures are made in Antofagasta region [4–6]. Moreover, the capture of individuals of this species is limited to five months per year, to protect the reproductive period in natural populations, which has generated great interest in aquaculture.

*Octopus mimus* is frequent in habitats of rocky shores of the Pacific Ocean in South America and it is distributed from Tumbes Bay in Northern Peru to San Vicente Bay in Central Chile [7]. Females have a unique oviposition during its life; eggs are laid in clusters and disposed on cave walls in the environment or on devices used as shelters in hatcheries. Eggs are incubated by the female, which constantly discards dead eggs to avoid the contamination of the healthy ones and remains taking care of them until the paralarvae hatch, approximately 80 days later. During the first weeks of life, paralarvae have planktonic habits capturing living prey as zoeas of crustacean and *Artemia*. Their diet during this early stage has been considered as the main critical factor for the survival of paralarvae in culture [7, 8].

In natural marine environments, invertebrates are colonized by microbial communities [9–11]. Increasing evidence suggests these communities are determined by host-selection, and provide beneficial functions including nutrient cycling, influence on morphogenesis and behavior, and protection from pathogens [12–14]. Nevertheless, how such specific microbial communities are assembled, maintained and which is their role in the survival during the critical life cycle stage, remains as central questions for understanding the ecological role of microorganisms in the host fitness. Identifying the ecological role of epibiotic bacteria-host interactions is challenging, since culture-dependent approaches don't allow them to fully characterize the community and may mask their behavior in the environment when growing isolated [11].

Previous studies report that bacteria associated with marine invertebrates provide beneficial traits, such as protection against pathogens and predators. As an example, the interaction between *Vibrio fischeri* and the squid *Euprymna scolopes*, demonstrated that bacteria give a luminescent appearance to the squid reducing the possibility of being preyed upon [15]. A highly specific relationship between microorganisms and host might be mediated by the production of antimicrobial compounds. In this context, it has been reported for the lobster *Homarus americanus* and the shrimp *Palaemon macrodactylus*, that Gram-negative epibiotic bacteria colonizing eggs provides protection to embryos through the production of antifungal compounds increasing their survival, there is evidence that antimicrobial peptides play a critical role in shaping the embryo microbiome and providing transgenerational protection [16–18].

Marine environments cover a vast surface of the world and support a great microbial diversity; hence constitute a promissory source of new potential natural products [19, 20]. Microorganisms living in symbiosis produce bioactive compounds as a response to the environment and host (e.g., physiology, age, among others), that could potentially aid in the maintenance of healthy tissue or individuals [11, 21]. In corals for example, an increase in water temperature induces changes in epibiotic bacterial communities, promoting growth of pathogens such as *Vibrio* and *Alteromonas* associated with diseases such as bleaching [22].

In *Octopus mimus*, as in other marine invertebrates, the big bottleneck for successful culture is the survival of paralarvae, as mortality is close to 90% during these development stages, as described for other closely related *Octopus* species [1, 23]. Mucus covering the surface of octopus offers a rich source of nutrients, especially for heterotrophic bacteria, and the competition

for space or nutrients between pathogenic and nonpathogenic species can promote healthy dynamics during early development stages in this species. In this context, there is evidence that fish mucus constitutes an active and dynamic barrier preventing the colonization of pathogens and infections [24]. Previously, several genes involved in secondary metabolites synthesis from bacteria *Lesingera* sp JC1 isolated from jelly coat on eggs of the bobtail squid (Cephalopoda) have been described [25], and it has been proposed that bacteria associated with early stages play an important role in protecting against predation, fouling, and pathogens in this invertebrate. Additionally, evidence suggests that other marine invertebrates are benefited by their associated microbiota, with great potential to produce bioactive compounds in nature and in culture farms [26, 27].

We hypothesized that characterization of the microbial community associated with different development stages of *O. mimus*, could contribute to understand the role of epibiotic microbes and the potential effects on survival of early development stages, where these first colonizers would avoid the colonization of pathogens by space competition or mediated by deleterious bioactive compounds. To our knowledge, microbiota associated with *O. mimus* is scarcely known; although it has been described that the eggs-associated bacteria are different depending on healthy and infected conditions [28]. Therefore, the main objective of this work was to describe the cultivable fraction of bacteria associated with mucus of male and female adults, paralarvae and eggs of *Octopus mimus*; and to evaluate the bioactive activities from these isolates, to contribute to the knowledge of the epibiotic bacterial communities colonizing different stages of life cycle of *O. mimus*.

## 2. Materials and methods

### 2.1 Samples collection and processing

This study analyzed the cultivable bacteria associated with eggs, paralarvae, and adults (female and male) of *Octopus mimus*. Adults were collected by divers in Caleta Bolsico (23˚ 28.454' S, 70˚ 37.328' W), a coastal zone of the Atacama Desert in Northern Chile. The water temperature at the sampling site was 15˚C, and salinity remained stable at 35 PSU throughout the year [29, 30], due to the absence of continental freshwater influence and low precipitation (approximately 7 mm year$^{-1}$).

The specimens (two males and one female) were transported to the laboratory in closed containers filled with seawater from the sampling sites. These containers included devices that served as refuges for the organisms to minimize transportation-induced stress. All organisms were subsequently maintained in the hatchery at the Laboratory of Larval Recirculation at Universidad de Antofagasta for additional studies. Immediately upon arrival at the laboratory, individuals were rinsed with filtered seawater (0.2 μm) to remove loose materials and debris. Mucus from an additional female incubating her eggs in the hatchery was sampled following the same procedure.

Eggs and paralarvae (1 month of age, 3 mm in length) obtained from the hatchery were used to prepare homogenized samples.

### 2.2 Isolation of microorganisms

Mucus samples were obtained using sterile cotton swabs, which were immediately submerged in marine saline solution (Oxoid) and streaked onto Zobell agar plates. Serial dilutions ($10^{-2}$ to $10^{-4}$) were prepared, and 100 μL aliquots were streaked onto Zobell agar plates. These plates were incubated at 20˚C with a 12:12 hour photoperiod for seven days until colony growth was observed. Enrichment culture of mucus was conducted in ST-10 medium (composition: 5 g·L$^{-1}$ peptone, 0.5 g·L$^{-1}$ yeast extract, 1 g·L$^{-1}$ glucose, 1 g·L$^{-1}$ iron (II) ammonium citrate, 15

g·L$^{-1}$ bacteriological agar), tryptic soy broth (TSB; Oxoid), and marine broth (MB; Difco, Co.) for 7 days at 20˚C without agitation. Subsequently, 100 μL aliquots were streaked onto Petri dishes containing tryptic soy agar (TSA; Oxoid, Co.), Zobell agar (Oxoid), Mueller-Hinton agar supplemented with 2.5% NaCl (Oxoid, Co.), or ST-10 agar culture media. All samples were incubated at 20˚C until colony growth was observed.

Distinct colony morphotypes were re-streaked onto their respective culture media to obtain axenic cultures. Isolated bacteria were cultivated in marine broth (Oxoid) at 20˚C to generate biomass for subsequent enzymatic and molecular assays. Bacterial isolates designated for enzymatic activity studies were preserved in 10% dimethyl sulfoxide (DMSO) and 30% glycerol; while samples for molecular analysis were stored in miliQ water at -20˚C until further use. All isolates were deposited in the Bacterial Collection at the Laboratory of Molecular Ecology and Applied Microbiology at Universidad Católica del Norte (Chile).

## 2.3 Molecular analysis and sequencing

Genomic DNA from pure cultures was extracted using the PowerSoil DNA Isolation Kit (MoBio) and stored at -20˚C. The 16S rRNA gene was amplified by polymerase chain reaction (PCR) using the universal primers 27F and 1492R, following a protocol previously described for this porpoise [31]. Sequencing services were provided by Macrogen Co. (Korea), and the resulting sequences of approximately 1500 base pairs were deposited in the GenBank database under accession numbers KX218274.1, KX218275.1, KX218317, KX218318, KX218282 and AY034144.1.

## 2.4 Enzymatic activity assays

Enzymatic activity assays were conducted using 31 bacterial isolates as species representatives, selected based on activities previously described by other authors [32–34]. To evaluate exoenzymatic activities, bacterial isolates were cultured at 20˚C on tryptic soy agar (TSA) media supplemented with 2.5% NaCl and various substrates as follows: a) Lipolytic activity: 1% Tween 20 + 0.001% CaCl₂, b) Proteolytic activity: 1% gelatin, c) Polysaccharolytic activity: 1% starch, d) Cellulolytic activity: 1% carboxymethylcellulose. DNase activity was assessed using DNase agar, while hemolytic activity was evaluated using blood agar supplemented with 5% sterile defibrinated sheep blood (BD Columbia Agar, Biomérieux). All enzymatic activities were evaluated after 48 hours of incubation at 20˚C.

## 2.5 Antibacterial activity assays

A modified double-layer Dopazo method [35] was employed to assess the antagonistic activities of all isolates against three pathogens of interest in aquaculture: *Vibrio parahaemolyticus* Kx RIMD2210633 (Provided by Dr. Romilio Espejo, Institute of Nutrition and Food Technology of the University of Chile, Chile), *V. anguillarum-related VAR* [36], and *V. ordalii* ATCC33509 (Provided by Dr Carlos Riquelme, Faculty of Marine Resources, Universidad de Antofagasta, Chile). Bacterial isolates were cultured in 5 mL of Marine Broth (Difco) media and incubated at 20˚C for 48 hours with agitation at 120 rpm. Subsequently, 10 μL of each culture was inoculated onto the center of Mueller-Hinton and marine agar plates, and colonies were grown for 48 hours at 20˚C under constant illumination.

Following incubation, colonies were exposed to chloroform vapors for 30 minutes. An overlay of soft agar (tryptic soy broth with 2.5% NaCl and 0.9% bacteriological agar) at 40˚C, previously inoculated with 100 μL of *Vibrio* isolates (grown overnight in tryptic soy broth at 37˚C), was then applied to the plates. The double-layer agar plates were incubated for 48 hours at 20˚C.

Antagonistic activities were evidenced by the presence of an inhibition halo around the colonies at the center of the plate [37]. Strain C33, a member of the *Vibrionaceae* family, was used as a positive control due to its known antimicrobial activity [36]. To determine antibacterial activities, the inhibition halos were measured in millimeters, where ≥10 mm is considered as strong inhibition [38].

## 2.6 The UHPLC analysis

For metabolomic analyses, strain M8-1 was selected due to its high inhibitory activity against pathogenic *Vibrio* in the antagonistic assay.

Bioactive compounds were extracted from 500 mL of bacterial culture in the stationary phase using ethyl acetate as a solvent in a 1:1 ratio to separate organic and aqueous phases. The aqueous phase was discarded, and the organic phase was mixed with HPLC-grade water (Merck) in a 1:2 v:v ratio. This mixture was then dried in a rotary evaporator at 60°C (Quickfit RE100). Residual moisture was removed by bubbling with nitrogen gas, and the resulting precipitate was resuspended in 500 μL of an ethanol:water (4:1 v:v) solution. This organic extract was used for subsequent experiments.

Liquid chromatography (LC) parameters were as follows: LC was performed using an UHPLC C18 column (Acclaim, 150 mm × 4.6 mm ID, 2.5 μm, Thermo Fisher Scientific, Bremen, Germany) operated at 25°C. The mobile phases consisted of 1% formic aqueous solution (A) and acetonitrile (B). The gradient program (time (min), % B) was: (0.00, 5); (5.00, 5); (10.00, 30); (15.00, 30); (20.00, 70); (25.00, 70); (35.00, 5), with a 12-minute column equilibration period before each injection. The flow rate was set at 1.00 mL min$^{-1}$, and the injection volume was 10 μL.

The previously reported Thermo Scientific Dionex Ultimate 3000 UHPLC system, hyphenated with a Thermo Q Exactive Focus machine, was utilized for the analysis. A 5 mg sample of the ethyl acetate extract from the strain *Bacillus megaterium* 8–1 (isolated from *O. mimus*) was dissolved in 2 mL of methanol, filtered through a PTFE filter, and 10 μL were injected into the instrument. All specifications were set as previously reported [39]. Identification was performed XCalibur 2.4 and Trace Finder 3.3 (Thermo Fisher Scientific, Bremen, Germany) using the exact molecular formula and MS/MS spectra and confirmed by the available literature. The MS/MS databases used were mzCloud (https://www.mzcloud.org/, accessed on 22 July 2024, Thermo Fisher Scientific), Mass Bank of North America (MoNA, http://mona.fiehnlab.ucdavis.edu, accessed on 22 July 2024), Global Natural Product Social Molecular Networking (GNPS, http://gnps.ucsd.edu, accessed on 22 July 2024), and Human Metabolome Database (HMBD, http://www.hmdb.ca/, accessed on 26 July 2024).

## 3. Results

In this study, a total of 49 bacterial isolates were obtained from different developmental stages of *O. mimus*, distributed as follows: 20 from eggs, 8 from paralarvae, 11 from adult female mucus, and 10 from adult male mucus (Table 1). The cultivable epibiotic bacteria were represented by 19 genera distributed across 4 phyla: 63.2% belonging to Proteobacteria, 21.1% to Actinomycetota, 10.5% to Bacillota, and 5.3% to Bacteroidota (Fig 1 and Table 1). Proteobacteria and Bacillota were found in all developmental stages; Actinomycetota was present only in adults (male and female), and Bacteroidota exclusively in paralarvae (Fig 1).

Bacterial enrichment cultures obtained from females demonstrated a high abundance of microorganisms living in the mucus layer over the skin (Fig 2). Bacterial isolates formed both pigmented and non-pigmented colonies when grown on agar (Fig 2A–2H), with some exhibiting elastic colonies belonging to the *Bacillus pumilus* species (Fig 2J). The highest specific

**Table 1. Identification of epibiotic bacterial isolates obtained from *Octopus mimus* for each development stage.**

| Bacteria Isolate | Phyla | Taxa (Closest type strain) | Fragment lengh (bp) | Q coverage (%) | Max identity (%) | Accession number (NCBI) |
|---|---|---|---|---|---|---|
| E1(3) | Gammaproteobacteria | *Oceanisphaera sp.* | 1374 | 99 | 97 | NR_133801 |
| E2(2) | Bacillota | *Bacillus pumilus* | 1382 | 100 | 99 | NR_112637 |
| E3(2) | Bacillota | *Bacillus megaterium* | 1414 | 100 | 99 | NR_112636 |
| E4(2) | Gammaproteobacteria | *Pseudoalteromonas carrageenovora* | 1376 | 100 | 99 | NR_113605 |
| E5(2) | Gammaproteobacteria | *Vibrio alfacsencis* | 1396 | 100 | 98 | NR_025479 |
| E6 | Gammaproteobacteria | *Shewanella algae* | 1370 | 100 | 99 | NR_117771 |
| E7 | Gammaproteobacteria | *Cobetia amphilecti* | 1369 | 100 | 100 | NR_113404 |
| E8 | Gammaproteobacteria | *Oceanisphaera donghaensis* | 1365 | 100 | 99 | NR_043622 |
| E9 | Gammaproteobacteria | *Shewanella algidipiscicola* | 1369 | 100 | 100 | NR_041297 |
| E10 (2) | Gammaproteobacteria | *Psychrobacter nivimaris* | 1379 | 100 | 99 | NR_028948 |
| E11 | Gammaproteobacteria | *Psychrobacter marincola* | 1351 | 100 | 99 | NR_025458 |
| E12 | Alphaproteobacteria | *Sphingobium yanoikuyae* | 1326 | 100 | 99 | NR_113730 |
| E13 | Gammaproteobacteria | *Vibrio neocaledonicus* | 1362 | 100 | 99 | JQ934829 |
| P1 | Alphaproteobacteria | *Mameliella atlantica* | 1297 | 99 | 100 | NR_136489 |
| P2 | Gammaproteobacteria | *Oceanisphaera donghaensis* | 1376 | 100 | 99 | NR_043622 |
| P3 | Alphaproteobacteria | *Sulfitobacter dubius* | 1310 | 99 | 99 | NR_025691 |
| P4(2) | Alphaproteobacteria | *Leisingera aquimarina* | 1306 | 99 | 98 | NR_042670 |
| P5(2) | Bacillota | *Bacillus cibi* | 1412 | 99 | 100 | NR_042974 |
| P6 | Bacteroidota | *Psychroserpens damuponensis* | 1339 | 100 | 99 | NR_109097 |
| F1(5) | Bacillota | *Bacillus pumilus* | 1377 | 100 | 99 | NR_112637 |
| F2 | Alphaproteobacteria | *Brevundimonas vesicularis* | 1302 | 100 | 99 | NR_113586 |
| F3 | Bacillota | *Staphylococcus warneri* | 1379 | 99 | 99 | NR_025922 |
| F4 | Actinomycetota | *Salinibacterium amurskyense* | 1354 | 100 | 99 | NR_041932 |
| F5 | Gammaproteobacteria | *Vibrio mytili* | 1397 | 100 | 99 | NR_044911 |
| F6 | Gammaproteobacteria | *Shewanella algae* | 1372 | 100 | 99 | NR_114236 |
| F7 | Actinomycetota | *Leifsonia aquatica* | 1328 | 100 | 99 | NR_119082 |
| M1 | Gammaproteobacteria | *Vibrio alginolyticus* | 1399 | 99 | 99 | NR_121709 |
| M2 | Gammaproteobacteria | *Vibrio neocaledonicus* | 1357 | 100 | 99 | JQ934829 |
| M3 | Gammaproteobacteria | *Pseudomonas geniculata* | 1382 | 100 | 99 | NR_024708 |
| M4 | Gammaproteobacteria | *Cobetia amphilecti* | 1376 | 99 | 99 | NR_113404 |
| M5 | Gammaproteobacteria | *Vibrio rumoiensis* | 1389 | 100 | 98 | NR_024680 |
| M6 | Actinomycetota | *Microbacterium esteraromaticum* | 1309 | 100 | 99 | NR_026468 |
| M7 | Bacillota | *Bacillus pumilus* | 1398 | 100 | 99 | NR_112637 |
| M8(2) | Bacillota | *Bacillus megaterium* | 1388 | 100 | 100 | NR_112636 |
| M9 | Actinomycetota | *Kocuria salsicia* | 1352 | 100 | 99 | NR_117299 |

(): numbers in parenthesis correspond to the number of isolates per taxa. E: eggs; P: paralarvae; F: female mucus; M: male mucus.

richness was observed in Proteobacteria, including 19 species, followed by Actinomycetota with 4 species, Bacillota with 4 species, and Bacteroidota with 1 species. *Vibrio* and *Bacillus* were the most abundant genera, with 7 and 14 isolates represented by 3 and 5 species, respectively. *Bacillus* was present in all developmental stages; while *Vibrio* was not isolated from paralarvae.

The culturable bacterial community colonizing the octopus differed depending on the developmental stage, and our results indicate that some taxa might be specific to each stage. For instance, eggs presented the highest bacterial richness with 8 genera and 14 species, where

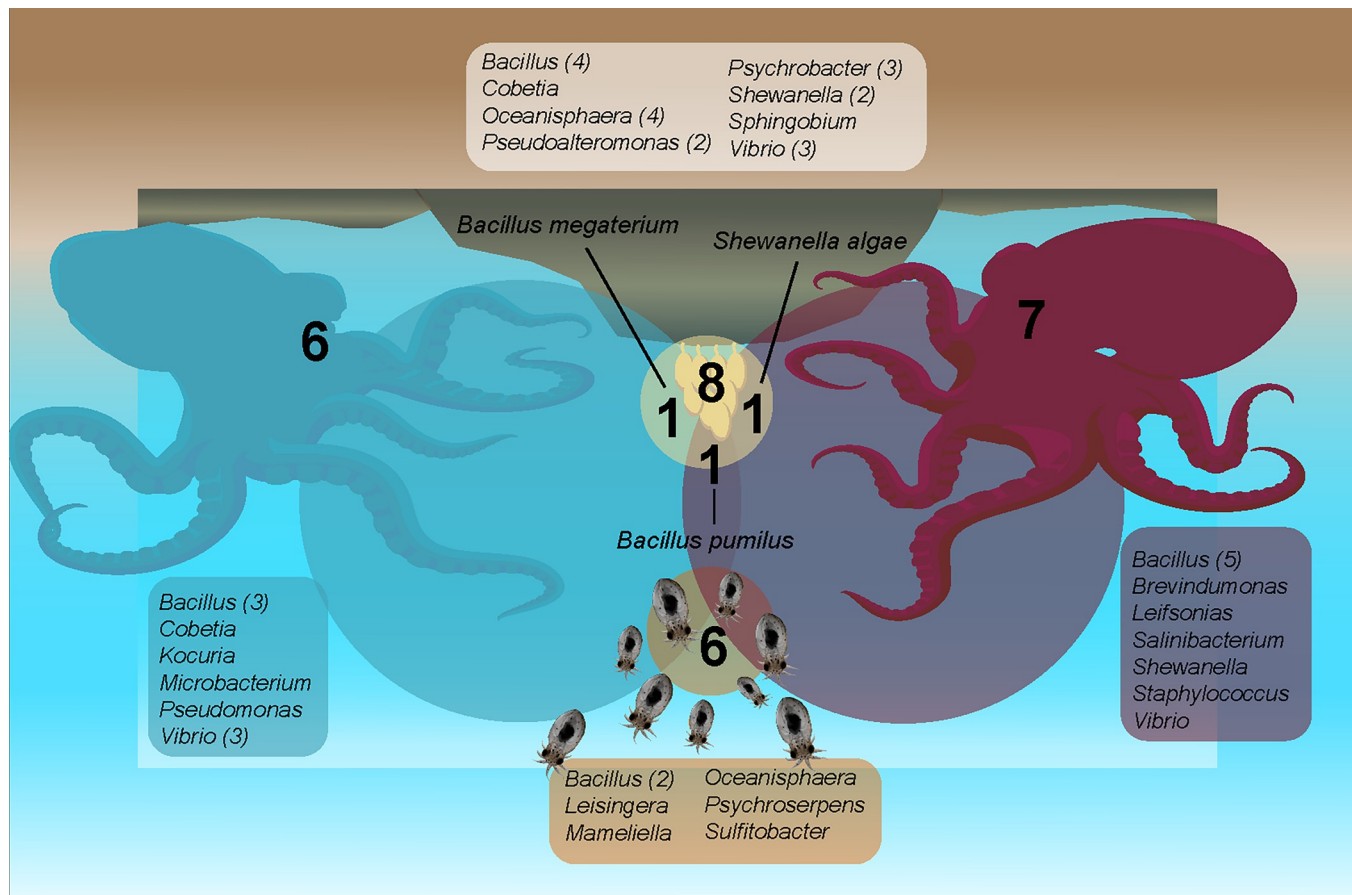

**Fig 1. Venn diagram for cultivable epibiotic bacterial phyla per development stages of *Octopus mimus*.** N indicates the number of genera for each development stage, and parentheses show the number of isolates per genus. Non-parenthesis indicates a unique isolate per genus.

*Sphingobium*, *Pseudoalteromonas*, and *Psychrobacter* genera were found exclusively in eggs. Paralarvae were colonized by 6 species distributed across 6 different genera, with four genera unique to this stage: three belonging to Roseobacteraceae (Alphaproteobacteria)—*Leisingera*, *Mameliella*, and *Sulfitobacter*—and one Bacteroidota belonging to the *Psychroserpens* genus (Fig 2).

Adults were colonized by phyla Proteobacteria, Actinomycetota, and Bacillota, which were represented by 12 genera and 15 species, with higher specific richness observed in males. Genera exclusive to males were *Kocuria*, *Microbacterium* (Actinomycetota), and *Pseudomonas* (Gammaproteobacteria) (Fig 2). Unique genera from females were *Brevundimonas*, *Staphylococcus*, and two Actinomycetota (Microbacteriaceae)—*Leifsonia* and *Salinibacterium*.

Few species were shared between different developmental stages. *Bacillus pumilus* was present in all samples except paralarvae; *Oceanisphaera donghaensis* was found in both eggs and paralarvae; *B. megaterium*, *Cobetia anfilecti*, and *Vibrio neocaldonicus* were shared by eggs and males, while *Shewanella algae* was present in both eggs and females (Table 1).

### 3.1 Antibacterial activities from epibiotic bacteria

All bacterial isolates obtained were evaluated for antibacterial activity against *Vibrio anguillarum*, *V. ordalii*, and *V. parahaemolyticus* (Table 2). *Bacillus pumilus* (E.2.1) and *B.*

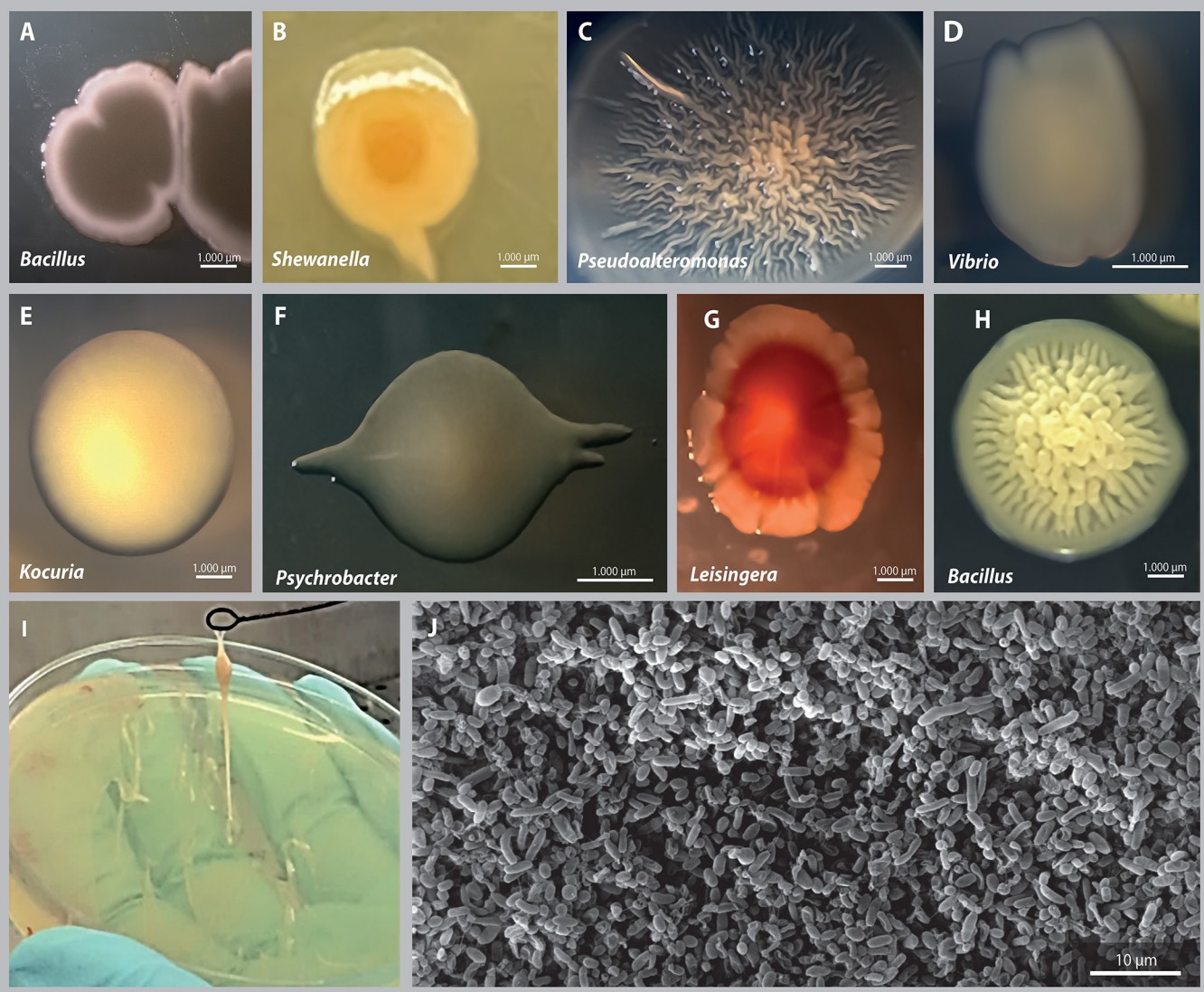

**Fig 2. Morphotypes of bacterial isolates from *Octopus mimus*.** A-H) Colonies of bacterial isolates, I) *Bacillus pumilus*, J) Scanning electron microscopy from mucus enrichment culture.

**Table 2. Inhibitory activities against pathogenic *Vibrio* species detected from epibiotic bacterial isolates.**

| Isolate | Species | Diameter of inhibition zone (mm) | | | Genbank accession code |
|---------|---------|------|------|------|------------------------|
| | | **Vp** | **Va** | **Vo** | |
| **E2.1** | *Bacillus pumilus* 1 | 19 | - | - | KX218274.1 |
| **E2.2** | *Bacillus pumilus* 2 | 29 [†] | 17 [†] | 20 [†] | KX218275.1 |
| **M8.1** | *Bacillus megaterium* 1 | 25 | 18* | 14 | KX218317 |
| **M8.2** | *Bacillus megaterium* 2 | 25 | - | - | KX218318 |
| **E6** | *Shewanella algae* | 30[†] | 32[†] | 30[†] | KX218282 |
| **C33** | *Vibrio* sp.3 | 27 | 14 | 20 | AY034144.1 |

Vp: *Vibrio parahaemolyticus*, Va: *V. anguillarum*, Vo: *V. ordalii*, (-): no inhibitory activity detected. [†] isolates with higher inhibitory activity compared to the C 33 reference strain.

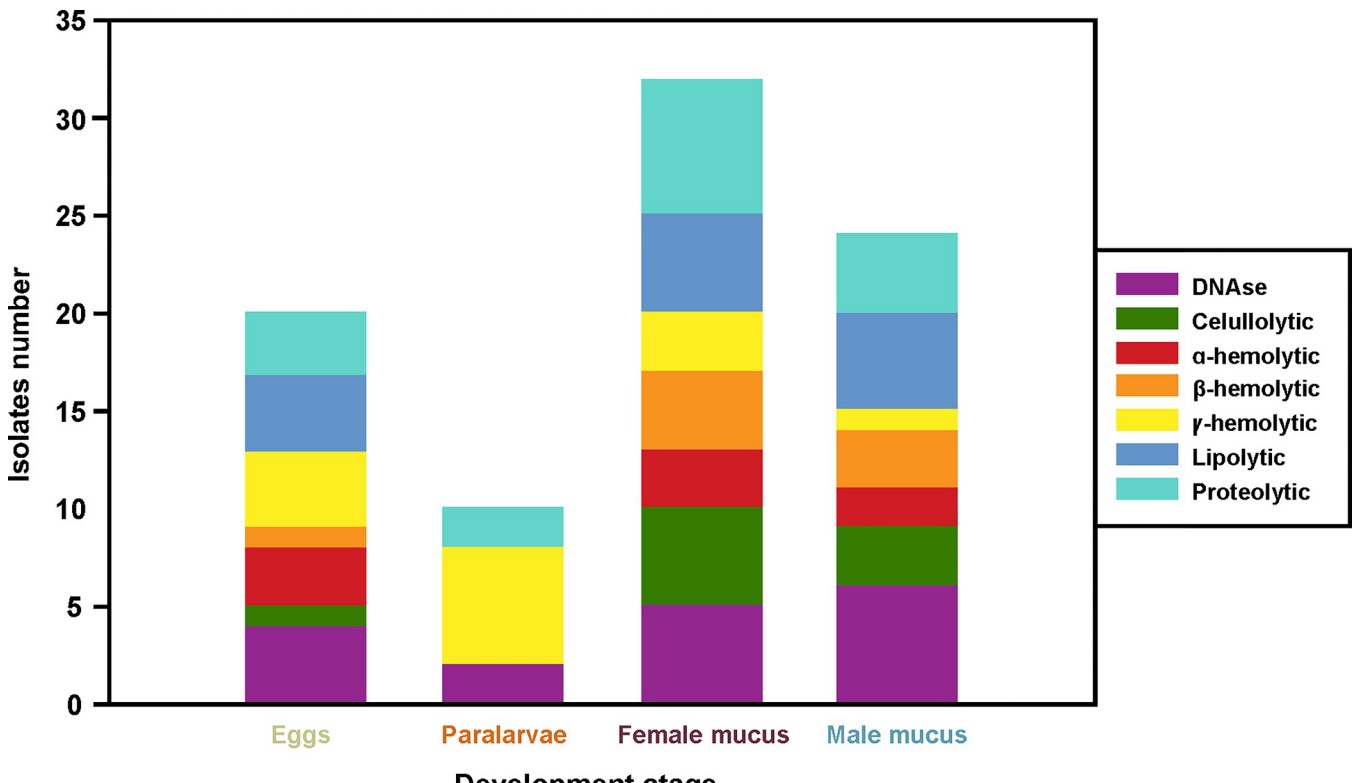

**Fig 3. Enzymatic activities assayed in bacterial isolates from *Octopus mimus*.** Numbers in bars represent the representative isolates positive for each activity. Every activity was assayed once, for each bacterial isolation.

*megaterium* (M8-2) exhibited antagonistic activity exclusively against *V. parahaemolyticus*. In contrast, *Shewanella algae* (E6), *B. pumilus* (E2-2), and *B. megaterium* (M8-1) demonstrated antagonistic activity against all three pathogens assayed (Table 2).

### 3.2 Enzymatic activities from epibiotic bacteria

Of the 31 bacterial isolates assayed, the most prevalent activity detected was DNase, exhibited by 56.7% of the isolates. Proteolytic activity was observed in 53.3% of isolates, lipolytic activity in 46.7%, cellulolytic activity in 30%, and hemolytic activity in 26.7% (Fig 3 and Table 3). No

**Table 3. Summary of enzymatic activities determined in epibiotic bacteria from *Octopus mimus*.**

| Bacterial Isolates | Blast | ENZYMATIC ACTIVITIES | | | | | |
|---|---|---|---|---|---|---|---|
| | (Closest type strain) | DNAse | Cellulolytic | α-hemolytic | β-hemolytic | Lipolytic | Proteolytic |
| E3, M8-1, M8-2 | *Bacillus megaterium* | + | + | - | + | + | + |
| F1-1, F1-2, F1-5, M7 | *Bacillus pumilus* | + | + | - | + | + | + |
| F1-4 | *Bacillus pumilus* | - | + | - | + | + | + |
| M9 | *Kokuria salsicia* | - | + | + | - | - | - |
| F7 | *Leifsonia aquatica* | + | - | - | - | - | - |
| P4-1, P4-2 | *Lesingera sp.* | - | - | - | - | - | + |
| P1 | *Mameliella atlantica* | + | - | - | - | - | - |
| P2 | *Oceanisphaera dongahensis* | + | - | - | - | - | - |
| E4-2 | *Pseudoalteromonas carrageenovora* | - | - | - | - | - | + |

agarolytic activity was detected. Proteolytic, lipolytic, cellulolytic, β-hemolytic, and DNase activities were more pronounced in bacterial isolates from male and female mucus. In contrast, bacterial isolates from paralarvae showed negative results for lipolytic, cellulolytic, and hemo- lytic (α and β) activities (Fig 3 and Table 3). The enzymatic activities demonstrated overall homogeneity at the genus level, particularly evident in *Bacillus*, *Vibrio*, and *Shewanella*, as illustrated in Table 3. Isolates also showed antimicrobial activity against *Vibrio* (Fig 4).

### 3.3 Bioactive compounds obtained by UHPLC analysis from *Bacillus megaterium* from males

Twenty compounds were detected in an ethyl acetate extract obtained from *Bacillus megater- ium* M8-1, isolated from male mucus, which was analyzed by UHPLC-MS (Fig 5). A compre- hensive metabolomics analysis was conducted, and mass spectra were obtained in both positive and negative modes (only negative mode data are presented) (Table 4). The analysis revealed the presence of nitrogenated compounds, bile acid-related compounds, oxylipins, gly- col derivatives, and other phenolics Fig 6 shows some representative compounds. The bioac- tive compounds identified include nitrogenated compounds, bile acids, oxylipins, glycol derivatives, and other phenolic compounds.

Nitrogenated compounds were detected, for example, Peak 2, with a $[M-H]^-$ ion at m/z 125.03455, was identified as thymine ($C_5H_5O_2N_2^-$). Peak 5, exhibiting a parent ion at m/z 215.08223 and a daughter ion at m/z 116.04977 ($C_8H_6N^-$ indole), was identified as lycopero- dine ($C_{12}H_{11}O_2N_2^-$), a constituent previously reported in the starfish *Asterias rollestoni* [40]. Peak 6 was identified as its derivative, 3-formylindole ($C_9H_6ON^-$). Another peak 6, with a $[M-H]^-$ ion at m/z 215.08223, was identified as 3-indoleacetic acid ($C_{10}H_6O_2N^-$) [41]. Peak 8 was identified as 3-hydroxyanthranilic acid methyl ester ($C_8H_8O_3N^-$) [42].

Additionally, Bile acids and related compounds were found, including Peak 13, with a pseu- domolecular ion at m/z 391.28561, was identified as murocholic acid ($C_{24}H_{39}O_4^-$) [43]. Peak 11 was identified as muricholic acid ($C_{24}H_{39}O_5^-$). Peak 12, exhibiting a pseudomolecular ion at m/z 389.26999, was identified as its derivative, dehydro-muricholic acid ($C_{24}H_{37}O_4^-$). Peak 14 was identified as ursodiol ($C_{24}H_{39}O_4^-$) [44], and peak 15 as dehydroursodiol ($C_{24}H_{37}O_4^-$), respectively.

On the other hand, Oxylipins correspond to Peak 9, with a $[M-H]^-$ ion at m/z 329.23349, was identified as trihydroxy-octadecaenoic acid ($C_{18}H_{33}O_5^-$). Peak 18, exhibiting a $[M-H]^-$ ion at m/z 257.21213, was identified as hydroxypentadecanoic acid ($C_{15}H_{29}O_3^-$). Both compounds are fatty acid oxylipin derivatives [45]. Also, Glycol derivatives such as Peak 16, with a $[M-H]^-$ ion at m/z 339.20010, was identified as heptaethylene glycol monomethyl ether ($C_{15}H_{31}O_8^-$). Peak 17 was identified as hexaethylene glycol dimethyl ether ($C_{15}H_{31}O_8^-$). Peak 19 was identi- fied as ethoxylated pentaerythritol ($C_{13}H_{27}O_8^-$).

Finally other phenolics compounds were identified, Peak 3, with a $[M-H]^-$ ion at m/z 238.09846, was identified as 4-(3-phenylpropyl)benzoic acid ($C_{16}H_{14}O_2^-$). Peak 4, exhibiting a $[M-H]^-$ ion at m/z 121.02880, was identified as benzoic acid ($C_7H_5O_2^-$). Peak 10, with a $[M-H]^-$ ion at m/z 405.26498, was identified as fukanefuromarin D ($C_{24}H_{27}O_5^-$) [46].

## 4. Discussion

Marine microorganisms are abundant in the ocean, reaching $10^4$ to $10^6$ cells mL$^{-1}$ in free living lifestyle [47], and bacterial communities colonizing marine invertebrates reach lower relative abundance than surrounding environment as described for corals [48, 49], sponges [50, 51], crustaceans [16, 52], tunicates [53], among others. This work reports for the first time the

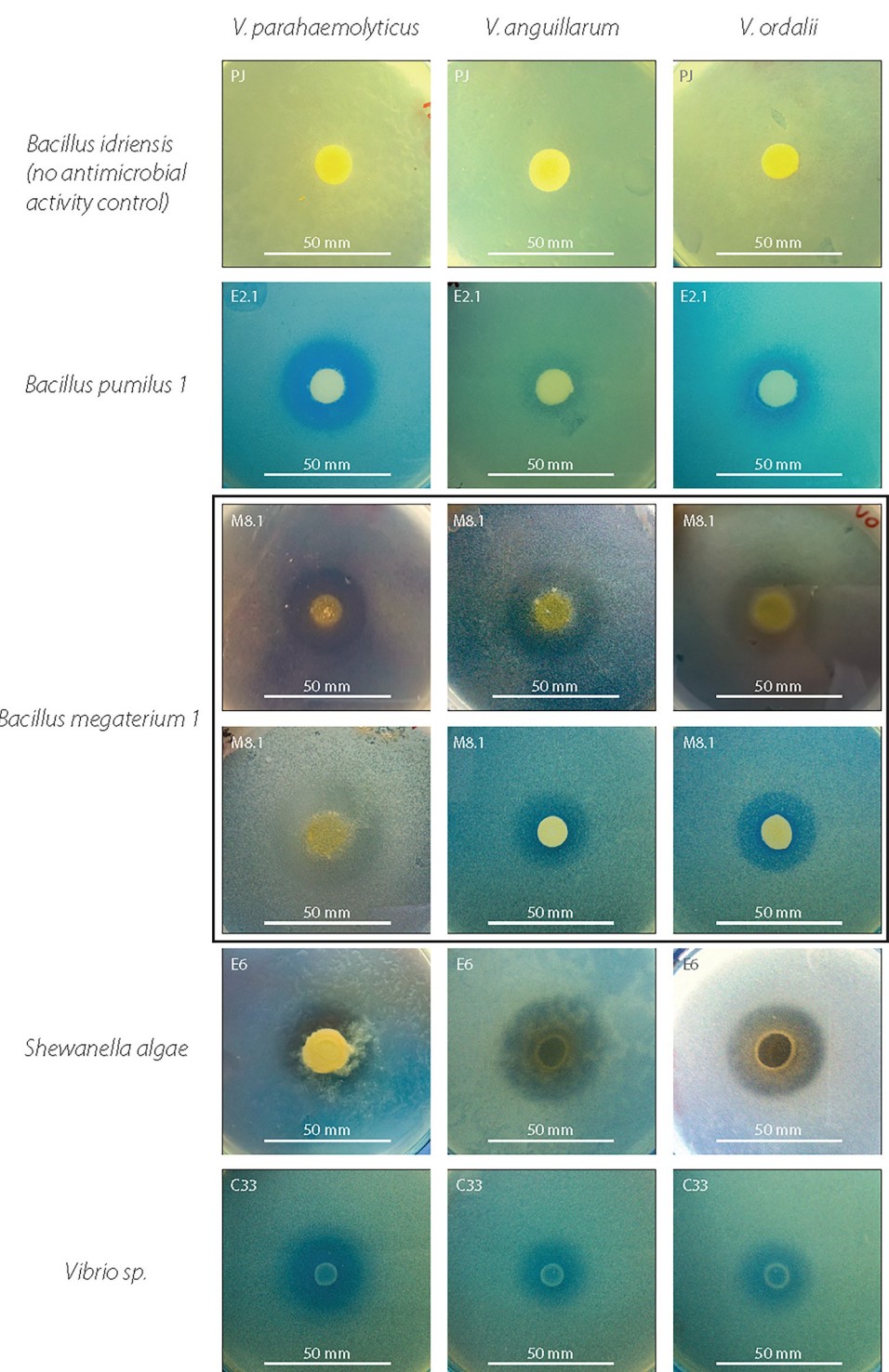

**Fig 4. Antibacterial activities of each isolate obtained from *Octopus mimus*.** Photographs show antibacterial assays against three marine pathogenic *Vibrio* species.

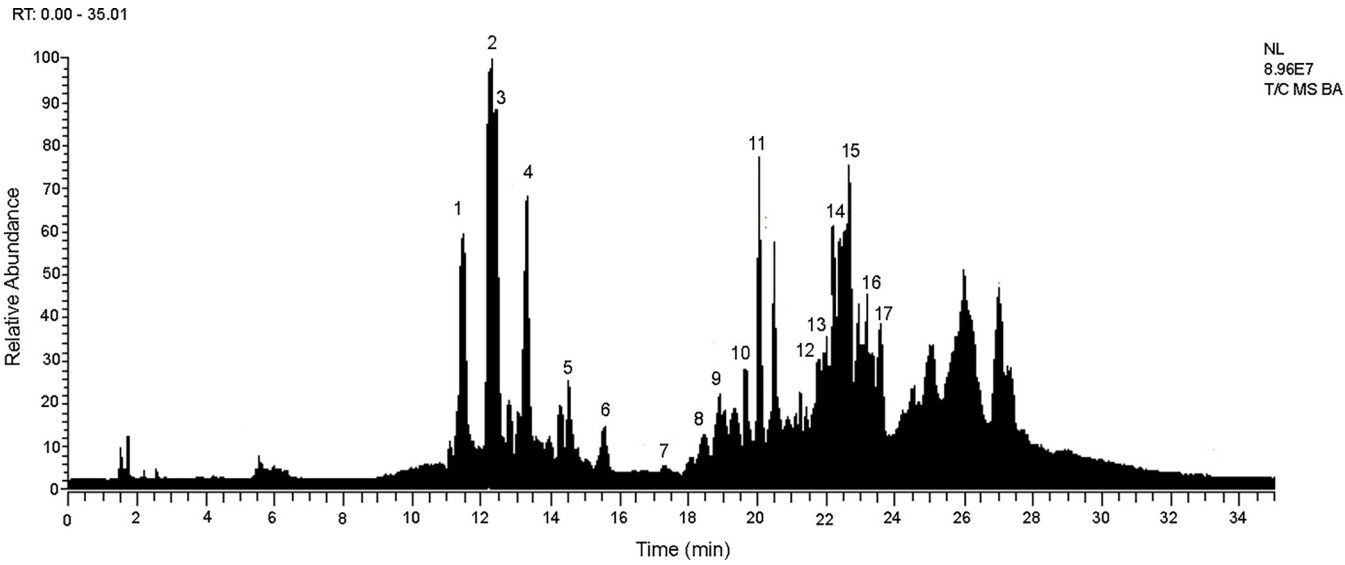

**Fig 5. UHPLC chromatograms of *Octopus mimus* extract.**

**Table 4. Fractionation of compounds from *Bacillus megaterium* obtained from mucus from *O. mimus*.**

| Peak # | Tentative identification | Elemental composition [M-H] | Retention time (min.) | Theoretical mass (m/z) | Measured mass (m/z) | Accuracy (Äppm) | MS$^n$ ions (Äppm) |
|---|---|---|---|---|---|---|---|
| 1 | Benzoic acid | $C_7H_5O_2^-$ | 11.45 | 121.02950 | 121.02860 | -7.44 | |
| 2 | Lycoperodine | $C_{12}H_{11}O_2N_2^-$ | 12.25 | 215.08260 | 215.08223 | -1.72 | 116.04977 ($C_8H_6N^-$), indole |
| 3 | 3-indolacetic acid | $C_{10}H_6O_2N^-$ | 12.38 | 174.05605 | 174.05559 | -2.64 | 116.04974 ($C_8H_6N^-$), indole |
| 4 | 3-formilindol | $C_9H_6ON^-$ | 13.27 | 144.04549 | 144.04483 | -4.58 | 116.04973 ($C_8H_6N^-$), indole |
| 5 | 3-hydroxyantranilic acid methyl ester | $C_8H_8O_3N^-$ | 15.55 | 166.05097 | 166.05043 | -3.25 | 124.03972 ($C_6H_6O_2N^-$) |
| 6 | Dihydroxyhexadecenoic acid | $C_{16}H_{29}O_4^-$ | 19.06 | 285.20713 | 285.20724 | +0.39 | |
| 7 | Dihydroxyheptadecaenoic acid | $C_{17}H_{31}O_4^-$ | 19.65 | 299.22299 | 299.22278 | -0.70 | |
| 8 | Hydroxymiristic acid | $C_{14}H_{27}O_3^-$ | 21.75 | 243.19657 | 243.19646 | -0.45 | |
| 9 | Hydroxypentadecanoic acid | $C_{15}H_{29}O_3^-$ | 22.23 | 257.21222 | 257.21222 | 0.00 | |
| 10 | dihydroxyhexadecanoic acid | $C_{16}H_{31}O_4^-$ | 22.32 | 287.22278 | 287.22290 | 0.42 | |
| 11 | hydroxyhexadecenoic acid | $C_{16}H_{29}O_3^-$ | 22.39 | 269.21222 | 269.21228 | 0.22 | |
| 12 | Hydroxypentadecanoic acid (isomer 9) | $C_{15}H_{29}O_3^-$ | 22.49 | 257.21222 | 257.21213 | -0.35 | |
| 13 | hydroxyhexadecenoic acid (isomer 11) | $C_{16}H_{29}O_3^-$ | 22.65 | 269.21228 | 269.21216 | -0.46 | |
| 14 | hydroxyhexadecadienoic acid | $C_{16}H_{27}O_3^-$ | 22.82 | 267.19657 | 267.19662 | 0.19 | |
| 15 | Hydroxytridecanoic acid | $C_{13}H_{25}O_3^-$ | 22.88 | 229.18092 | 229.18071 | -0.92 | |
| 16 | dihydroxyheptadecanoic acid | $C_{17}H_{33}O_4^-$ | 22.90 | 301.23843 | 301.23856 | 0.43 | |
| 17 | hydroxyheptadecaenoic acid | $C_{17}H_{31}O_3^-$ | 23.20 | 283.22787 | 283.22797 | 0.35 | |

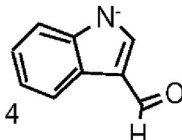

**Fig 6. Structures of some compounds detected in *Octopus* mucus.**

cultivable bacteria associated with all developmental stages of Chilean northern octopus, and our results suggest a tendency for bacteria associated with specific development stages.

In males two unique genera, not observed in other samples, were detected: *Kokuria* and *Microbacterium*, both belong to phylum Actinomycetota. This phylum has a high relevance in new bioactive compounds production, due to there are close to 22,000 secondary metabolites, of which 70% are from Actinomycetota and 7% belong to Phylum Bacillota from *Bacillus* species [54–57]; therefore, these are promising groups for finding new drugs from marine environments. In this study, 4 out of the 5 isolates that presented antagonistic activities against the marine pathogens tested corresponded to *Bacillus* species (2 isolates from male and 2 isolates from eggs).

Previous reports describe that epibiotic bacteria might have beneficial, neutral or detrimental effects on host fitness [27, 58–60]. Furthermore, bacterial competition for space or nutrients, avoid the colonization of other organisms potentially deleterious and protect the host; therefore, the first colonizers could be determinants in the community structure and survival of the host [12, 15, 61]. Early development stages of invertebrates are the most sensitive to negative effects of abiotic and biotic factors from the surrounding environment; for which the organisms have evolved several protective mechanisms. Parental protection mediated by antimicrobial peptides have been described; although bacterial bioactive producers with antimicrobial properties or also probiotics, seems to be the more frequent strategy, even used in the aquaculture industry [18, 27].

The main question related to egg survival from *O. mimus* is: Why might the eggs be contaminated after being laid by the female?. In this work, healthy eggs and females showed the highest diversity of bacteria, although only one *Shewanella* species was common among the samples. *Shewanella algae* is a rod-shaped Gram-negative marine bacterium mainly associated with pathogenesis; nevertheless, has been reported a strong inhibition of volatiles antifungal compounds against pathogenic fungi and a high inhibition against bacteria as *Vibrio ordalii* and *V. harveyi* [62, 63], two frequent pathogenic vibrios affecting marine invertebrates in the natural environment or in farming.

These findings contribute to the understanding of the relationship between *Octopus mimus* and its associated microbiota during its life cycle, as eggs and paralarvae stages are the bottleneck to a successful reproductive cycle and thus are relevant in the aquaculture industry. Uncultured bacterial communities from eggs, revealed that Roseobacter clade (Alphaproteobacteria) bacteria are dominant in healthy eggs, and other Proteobacteria (*Pseudoalteromonas*, *Shewanella*, *Vibrio*) is found in infected eggs of *O. mimus*. Proteobacteria is the dominant phylum among the analyzed samples, particularly members of Rhodobacteraceae family were present in paralarvae, namely representatives from the genera *Leisingera*, *Sulfitobacter*, and *Mameliella*, suggesting a healthy condition for paralarvae [28]. Females from many cephalopod species contain *Leisingera* species on accessory nidamental glands, being transferred to eggs and giving them protection from predators or pathogens [25]. Likewise, several bioactivities are associated with *Sulfitobacter* species including channel-blocking toxins and cyclopeptides as diketopiperazines [64]. Cyclopeptides are antimicrobial compounds which can act as quorum sensing sensors in cross-talking microbial interactions [22, 64], which can inhibit host colonization.

On the other hand, bacteria living on invertebrates are producing chemical molecules acting as a deterrent for new colonizers, in particular marine environments have attracted the interest of researchers as a vast source of new biological molecules with bioactive properties [65, 66]. Antimicrobial compounds from marine bacteria have been reported from many species of *Bacillus*, such as *B. pumilus* against *V. alginolyticus*, *V. harveyi*, *V. mimicus*, *V. parahaemolyticus* and *V. cholera* [67–69], and against Gram positive bacteria like *Arthrobacter citreus* [70] by action of molecules as diketopiperazines [25, 70, 71]. Our results demonstrated that *B. pumilus* and *B. megaterium* also inhibited the growth of pathogenic species such as *V. anguillarum* and *V. ordalii*, which has not been previously reported.

*Shewanella algae* inhibits the growth of *V. parahaemolyticus* and *V. alginolyticus* species [72] but does not affect *V. anguillarum* and *V. ordalii*. In this work, *Shewanella algae* was isolated from both, eggs and female mucus, moreover the isolate *S. algae* E6 from eggs, presented antagonistic activity against the three pathogens tested, suggesting a protective role of bacteria from eggs preventing the colonization by other bacteria, different from the pathogenic role regularly associated to *S. algae* [73]. Beneficial action of *S. algae* and other species belonging to *Shewanella* genus were reported previously associated with accessory nidamental glands (ANG) and eggs capsule of the squid *Loligo pealeii* [74].

In marine environments, *Bacillus* species are frequent colonizers of biotic surfaces, and our results report different species in all developmental stages. *Bacillus pumilus* isolated from eggs and *B. megaterium* from male mucus, showed antagonistic activities against pathogenic *Vibrio*. It has also been described that *B. megaterium* synthesizes bioactive compounds inhibiting the growth of other microorganisms [75]. Particularly in our study, the bioactive compounds from *B. megaterium* M8-1 isolated from males was analyzed by UHPLC. Results demonstrated that it produces seventeen bioactive compounds that belong to putative phenolic acids, indole derivatives and oxylipins, based on predicted chemical structure. Phenolic acids are a subclass of phenolic compounds (one of the most diverse groups of secondary metabolites, that have aromatic rings attached to one or more functional groups) described with antimicrobial activity and antifungal activity [76]. In addition, indole derivatives are known to display various bioactivities such antimicrobial, antiviral and antiparasitic activities [77–79] its synthesis has been described by marine sponges [80, 81] and by marine microorganisms [82]. Both phenolic acids and indole derivatives could be useful in the different stages of *O. mimus* development, contributing to the inhibition of pathogens mediated by the bioactive compounds produced by *B. megaterium*.

Also, oxylipins are secondary metabolites that can disrupt the reproductive cycle, by affecting oocyte maturation; fertilization; embryogenesis, larval competence and developmental processes [83, 84]; while in the adult males it would not produce negative effects.

Overall, our results contribute to better understanding the role of epibiotic bacteria colonizing in different life cycle stages of Chilean northern octopus and, we demonstrated that there are specific cultivable epibiotic bacteria in each developmental stage. Some of these microorganisms have antibacterial activities against generalist pathogenic *Vibrio* species, frequent in marine environments and of economic relevance for aquaculture. Additionally, extracellular bacterial enzymes may be both beneficial and deleterious for host survival. Beneficial activities have been associated with cellulolytic, lipolytic, and proteolytic activities [85, 86]; while DNase and hemolytic activities have been associated with pathogens for eukaryotes [87]. Regarding the isolated bacteria in this study, bacteria producing DNAse activities might be inhibiting growth of opportunistic pathogenic microorganisms in *O. mimus*.

Lipolytic activity is associated with bacterial lipases biosynthesis. These enzymes are widely produced in nature and preferentially hydrolyze substrates with long-chain fatty to short-chain fatty acids [85]. A beneficial function of lipases is that they can hydrolyze insoluble compounds to soluble ones in an aqueous environment such as seawater, and their chemical characteristics facilitate interaction between aqueous and non-aqueous interface [85], such as the mucosal surface of *O. mimus* and the environment.

Although culture of microorganisms has several limitations in representativeness in the octopus epibiont community, it remains the most valid strategy to identify bacteria producing antimicrobial compounds. The microbiome underpins the basis of animal health and is involved in virtually all metabolic processes; therefore, further studies are required to elucidate the complete composition of the epibiotic community of *O. mimus* and their ecological role on this interspecific relationship. Nowadays, high throughput sequencing approaches offer a valuable tool, revealing the complexity of microbial communities and exploring the bioactive potential of different organisms as never before, which would allow deeper characterization of the microbial-host dynamics.

## 5. Conclusion

*Octopus mimus* are colonized by different microorganisms during all development stages, and this switch in epibiotic bacteria composition might contribute to success in its life cycle. Here we evidenced that bacteria associated with different development stages, synthesize exoenzymes with activity that could modulate colonization by bacteria over time. Particularly, epibiotic bacteria that belong to the *Bacillus* genus, produce bioactive compounds with growth inhibitory activity; such as putative phenolic acids and indole derivatives synthesized by *B. megaterium*. These molecules could have a protective function against colonization, through space competition or growth inhibition by bioactive compounds with antibacterial properties. Further efforts are needed to characterize the microbiome of *Octopus mimus* by high-throughput sequencing to help decipher the role of microorganisms in the survival of the most critical stages of the Changos' octopus development.

## Acknowledgments

The author thanks to the Larval Recirculation Laboratory. Thanks to Mauricio Cáceres and Ricardo Utreras for providing the samples from the field. Thanks to Esteban Severino for the graphic edition and Pablo Paquis for the contribution to the final edition of the manuscript. Thanks, Dr. Carlos Riquelme for the laboratory facilities.

## Author Contributions

**Conceptualization:** Martha B. Hengst.

**Data curation:** Martha B. Hengst, Stephanie Trench, Valezka Alcayaga, Cristian Sepúlveda-Muñoz.

**Formal analysis:** Martha B. Hengst.

**Funding acquisition:** Martha B. Hengst.

**Investigation:** Martha B. Hengst, Stephanie Trench, Valezka Alcayaga, Cristian Sepúlveda-Muñoz, Jorge Bórquez, Mario Simirgiotis.

**Methodology:** Martha B. Hengst, Stephanie Trench, Valezka Alcayaga, Cristian Sepúlveda-Muñoz.

**Project administration:** Martha B. Hengst.

**Resources:** Martha B. Hengst.

**Supervision:** Martha B. Hengst.

**Validation:** Martha B. Hengst.

**Visualization:** Martha B. Hengst, Fernando Valenzuela, Mario Lody, Lenka Kurte, Coral Pardo-Esté.

**Writing – original draft:** Martha B. Hengst, Stephanie Trench, Lenka Kurte, Coral Pardo-Esté.

**Writing – review & editing:** Martha B. Hengst, Lenka Kurte, Coral Pardo-Esté.

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
