## [Decision Letter · Decision Letter 0]

30 Aug 2024

PONE-D-24-34526Epibiotic bacterial community composition varies during different developmental stages of Octopus minus: study of cultivable representatives and their secondary metabolite production.PLOS ONE

Dear Dr. HENGST,

Thank you for submitting your manuscript to PLOS ONE. After careful consideration, we feel that it has merit but does not fully meet PLOS ONE’s publication criteria as it currently stands. Therefore, we invite you to submit a revised version of the manuscript that addresses the points raised during the review process.

**ACADEMIC EDITOR: **The reviewers have commented about the MS and they found it quite interesting. However, both reviewers are agreed that the paper needs to be remarkable improved before considering for publication. Therefore, I recommend a major revision following closely all comments done by the reviewers.

We look forward to receiving your revised manuscript.

Kind regards,

Estibaliz Sansinenea

Academic Editor

PLOS ONE

Journal requirements: 1. When submitting your revision, we need you to address these additional requirements. Please ensure that your manuscript meets PLOS ONE's style requirements, including those for file naming. The PLOS ONE style templates can be found at https://journals.plos.org/plosone/s/file?id=wjVg/PLOSOne_formatting_sample_main_body.pdf and https://journals.plos.org/plosone/s/file?id=ba62/PLOSOne_formatting_sample_title_authors_affiliations.pdf. 2. Thank you for stating the following financial disclosure:  [CODEI 5388 Project UA (MB Hengst)].  Please state what role the funders took in the study.  If the funders had no role, please state: ""The funders had no role in study design, data collection and analysis, decision to publish, or preparation of the manuscript."" If this statement is not correct you must amend it as needed. Please include this amended Role of Funder statement in your cover letter; we will change the online submission form on your behalf. 3. Thank you for stating the following in the Acknowledgments Section of your manuscript: [This work was supported by CODEI 5388 Project of Universidad de Antofagasta. The author also thanks Larval Recirculation Laboratory, Mr. Mario Lody for providing the eggs and paralarvae. Thanks to Fernando Valenzuela, Mauricio Cáceres and Ricardo Utreras for providing the samples from the field, and Rubén Araya for pathogenic Vibrio strains for antimicrobial assays. Thanks, Dr. Carlos Riquelme for laboratory facilities. ]We note that you have provided funding information that is not currently declared in your Funding Statement. However, funding information should not appear in the Acknowledgments section or other areas of your manuscript. We will only publish funding information present in the Funding Statement section of the online submission form. Please remove any funding-related text from the manuscript and let us know how you would like to update your Funding Statement. Currently, your Funding Statement reads as follows:   [CODEI 5388 Project UA (MB Hengst)].   Please include your amended statements within your cover letter; we will change the online submission form on your behalf.

Additional Editor Comments:

The reviewers have commented about the MS and they found it quite interesting. However, both reviewers are agreed that the paper needs to be remarkable improved before considering for publication. Therefore, I recommend a major revision following closely all comments done by the reviewers.

Reviewers' comments:

Reviewer's Responses to Questions

**Comments to the Author**

1. Is the manuscript technically sound, and do the data support the conclusions?

Reviewer #1: Yes

Reviewer #2: Partly

2. Has the statistical analysis been performed appropriately and rigorously? 

Reviewer #1: No

Reviewer #2: N/A

3. Have the authors made all data underlying the findings in their manuscript fully available?

Reviewer #1: Yes

Reviewer #2: Yes

4. Is the manuscript presented in an intelligible fashion and written in standard English?

Reviewer #1: Yes

Reviewer #2: Yes

5. Review Comments to the Author

Reviewer #1: Review

The manuscript “Epibiotic bacterial community composition varies during different developmental stages of Octopus minus: study of cultivable representatives and their secondary metabolite production.” is well written and has interesting information for marine microbiologists, however, I found some errors and one figure of UHPLC-MC is missing. Please see the comments below:

Line 60-61. Please use just the most relevant references and writing: “deeply discussed in [13-16]” in the expression “are colonized by microbial communities [9-12], deeply discussed in [13-16].” is strange. This is not a review paper but you mentioned here in total eight references from 9 to 16. Choose just 2-3 most relevant references from 9 to 16 and that will be enough.

Line 205. Where is the statistical analysis? Did you analyze the results statistically? Please add section “3.7 Statistical analysis” to Methods.

Table 2. It is better to change the “inhibition hale” in “Diameter of inhibition hale (mm)” to “inhibition zone”.

Line 220. In the table 1, you showed the identification of bacterial isolates, however you should know that the majority of journals require registering the isolates in some database like GenBank of NCBI after identification and getting accession numbers for each isolate. However, you didn’t register them in GenBank or elsewhere.

Lines 279-285. Why did you put these methods into Results. All the methods should be in Methods section.

Line 286. Where are the results in the form of table or figure? Where can I see those twenty compounds? You need to show the results of UHPLC-MS and insert the reference to figure in the text.

Line 294. “4.4 Nitrogenated compounds, 4.5 Bile acids and related compounds, 4.6 Oxylipins, 4.7 Glycol derivatives, 4.8 Other phenolics” are all related to heading “4.3 Bioactive compounds obtained by UHPLC analysis from Bacillus megaterium from males”. So, the subheadings 4.4-4.8 should be joined as one text without numbers and can just be in italics to underline their position and relation to one common heading 4.3.

Line 295. “Peak 2, with a [M-H]⁻ ion at m/z 125.03455, was identified as thymine (C₅H₅O₂N₂⁻).” Where can we see this peak? Need to insert table of figure with results and reference to that fig/table in the text. This also concerns to subheading 4.5-4.8 where you talk about peaks but do not show the figure with peaks.

The resolution of tables 1 and 2 is very low and it is hard to see the details.

I recommend adding Figure S1 into manuscript as one of the ordinary figures but not supplementary, because it shows us the appearance of bacterial colonies and in general manuscript will look more presentable for the readers. The same related to Table S1. I recommend replacing it to main text as it would be more interesting to see all in the main text but not as supplementary.

Reviewer #2: The paper reported the epibiotic bacterial community composition cultivated from different developmental stages of Octopus minus and their secondary metabolite production. From my point of view, the paper needs to be remarkable improved before considering for publication.

The most important concern is that using a cultivation-dependent method for investigation of the epibiotic bacterial community composition cultivated from different developmental stages of Octopus minus is not convinced. The cultivation-dependent approaches are normally influenced by culture media and conditions more than their hosts. Also, the number of bacteria is not significant enough to draw a conclusion. Next-generation sequencing is available and used widely now in microbial diversity investigations. Why did authors not use cultivation-independent approaches for this section?

Provide sample collection time and who identified the Octopus mimus.

Lines 122-123: Salinity remained stable at 35 g/L throughout the year? How did you know that? How many times did you measure?

Lines 132-133: “Mucus samples were obtained using sterile cotton swabs, which were immediately submerged in marine saline solution (Oxoid) and streaked onto Zobell agar plates” should move to Section 3.2. Isolation of microorganisms

Lines 136-138: “Seriel dillution...” should move to Section 3.2. Isolation of Microorganisms

Line 174: Antimicrobial should be changed to Antibacterial

Provide the strain code of pathogenic strains Vibrio, and also their sources, collection, etc.

Why did authors not use microdilution for antibacterial assay? Agar-based assays may not be as exact as microdilution assays. Also, it will be better if authors use ethyl acetate extract for both antibacterial assay and UHPLC analysis.

Provide more information about UHPLC analysis. The authors only mentioned how to run the equipment but did not mention how to identify chemical compounds.

I suggest that the bacterial strains should only be identified at the genus level. It is not reliable to identify them as species level only based on one gene marker.

Provide the main figures of inhibition halo in antibacterial assays.

Why did the authors select the strain M8.1 for secondary metabolite analysis by UHPLC? Antibacterial activity of other strains such as E2.2 and E6 is better than M8.1.

Provide the figure of the UHPLC chromatogram.

Sections 4.4 to 4.8 should be organized as a table with columns such as a list of compounds, their retention time, m/z, references, etc.

Conclusion is vague and does not reflect the paper’s main results, thereby needing to be rewritten.

Please check the originality of image G in Figure S1. It seems that the size bar is not from the image but added.

6. PLOS authors have the option to publish the peer review history of their article (what does this mean?). If published, this will include your full peer review and any attached files.

Reviewer #1: **Yes: **Vyacheslav Shurigin

Reviewer #2: No

---

## [Author Response · Author response to Decision Letter 0]

15 Oct 2024

The responses to the reviewers are in the respective document attached to this reviewed submission by the name of "response to reviewers".

Please, find the same content below:

Comments to the Author

1. Is the manuscript technically sound, and do the data support the conclusions?

Reviewer #1: Yes

Reviewer #2: Partly

2. Has the statistical analysis been performed appropriately and rigorously?

Reviewer #1: No

Reviewer #2: N/A

3. Have the authors made all data underlying the findings in their manuscript fully available?

Reviewer #1: Yes

Reviewer #2: Yes

4. Is the manuscript presented in an intelligible fashion and written in standard English?

Reviewer #1: Yes

Reviewer #2: Yes

5. Review Comments to the Author

Reviewer #1: Review

The manuscript “Epibiotic bacterial community composition varies during different developmental stages of Octopus minus: study of cultivable representatives and their secondary metabolite production.” is well written and has interesting information for marine microbiologists, however, I found some errors and one figure of UHPLC-MC is missing. Please see the comments below:

We thank the reviewer for the comments that improved the manuscript, we have taken into consideration each raised concerns and have included new information to clarify. In particular, Figure 5 shows the UHPLC chromatograms of Octopus mimus extract.

Line 60-61. Please use just the most relevant references and writing: “deeply discussed in [13-16]” in the expression “are colonized by microbial communities [9-12], deeply discussed in [13-16].” is strange. This is not a review paper but you mentioned here in total eight references from 9 to 16. Choose just 2-3 most relevant references from 9 to 16 and that will be enough.

We thank the reviewer for the suggestion, we have implemented it and changes were made in the revised manuscript

Line 205. Where is the statistical analysis? Did you analyze the results statistically? Please add section “3.7 Statistical analysis” to Methods.

Due to the type of results, statistical analysis was not performed for the evaluation of enzymatic and antimicrobial activities. For the screening tha analysis was realized once per each isolate, except for B. megaterium 8.1, which was performed twice. A paragraph has been included in the methods section.

Table 2. It is better to change the “inhibition hale” in “Diameter of inhibition hale (mm)” to “inhibition zone”.

We changed the line in Table 2.

Line 220. In the table 1, you showed the identification of bacterial isolates, however you should know that the majority of journals require registering the isolates in some database like GenBank of NCBI after identification and getting accession numbers for each isolate. However, you didn’t register them in GenBank or elsewhere.

We have included the Genbank accession codes to each strain in Table 1. 

Lines 279-285. Why did you put these methods into Results. All the methods should be in Methods section.

We moved this section to Methods section, lines 136-140.

Line 286. Where are the results in the form of table or figure? Where can I see those twenty compounds? You need to show the results of UHPLC-MS and insert the reference to figure in the text.

Figure 5 was included as well as additional data in Table 4 and Figure 6.

Line 294. “4.4 Nitrogenated compounds, 4.5 Bile acids and related compounds, 4.6 Oxylipins, 4.7 Glycol derivatives, 4.8 Other phenolics” are all related to heading “4.3 Bioactive compounds obtained by UHPLC analysis from Bacillus megaterium from males”. So, the subheadings 4.4-4.8 should be joined as one text without numbers and can just be in italics to underline their position and relation to one common heading 4.3.

We thank the reviewer for the suggestion, we have adjusted the text in the revised version of the manuscript.

Line 295. “Peak 2, with a [M-H]⁻ ion at m/z 125.03455, was identified as thymine (C₅H₅O₂N₂⁻).” Where can we see this peak? Need to insert table of figure with results and reference to that fig/table in the text. This also concerns to subheading 4.5-4.8 where you talk about peaks but do not show the figure with peaks.

We have included the requested information in Figure 5 and Table 4

The resolution of tables 1 and 2 is very low and it is hard to see the details.

We thank the reviewer for this observation, we have increased the quality of the tables.

I recommend adding Figure S1 into manuscript as one of the ordinary figures but not supplementary, because it shows us the appearance of bacterial colonies and in general manuscript will look more presentable for the readers. The same related to Table S1. I recommend replacing it to main text as it would be more interesting to see all in the main text but not as supplementary.

We thank the reviewer for the suggestion, we have included this information in the main text as Figure 2 and Table 3.

Reviewer #2: The paper reported the epibiotic bacterial community composition cultivated from different developmental stages of Octopus minus and their secondary metabolite production. From my point of view, the paper needs to be remarkable improved before considering for publication.

The most important concern is that using a cultivation-dependent method for investigation of the epibiotic bacterial community composition cultivated from different developmental stages of Octopus minus is not convinced. The cultivation-dependent approaches are normally influenced by culture media and conditions more than their hosts. Also, the number of bacteria is not significant enough to draw a conclusion.

We thank the reviewer for this interesting suggestion, and we agree with the reviewer about the selectivity of culture media and total number of bacterial isolates. Nevertheless, even though the use of molecular techniques for microbial community analysis is an excellent tool to determine biological diversity (e.g., 16S rRNA, Illumina); the use of culture-dependent techniques is still an appropriate approach to determine antimicrobial properties or other biological activities in bacteria as we show in this work, as the activity is evidenced in vitro. One of the main advantages of culture-dependent methods is the lower cost, particularly for an initial analysis. However, the use of combined techniques such as UHPLC-Ms allowed us to determine that the M8.1 isolate synthesizes compounds with antibacterial activity that could be of interest.

Our results are not intended to be conclusive regarding the totality of bacteria living in association with Octopus mimus. This work provides new knowledge of the microbial community of this species, and particularly which of the bacteria identified here could constitute a good candidate for the search for bioactive molecules inhibitory to marine pathogens such as the representatives of the genus Vibrio.

To our knowledge, no other study published to date has described bacteria associated with all developmental stages of octopus, in particular Octopus mimus, a species native to the eastern Pacific ocean, and for which no commercial hatchery culture has so far been developed.

Next-generation sequencing is available and used widely now in microbial diversity investigations. Why did authors not use cultivation-independent approaches for this section?

Yes, we agree with the reviewer. NGS is a powerful technique to identify microbiota from any species or environment and has a great predictive value on gene clusters involved in the biosynthesis of antimicrobial bioactives (e.g., BGCs, PKS, NRPS, RiipS, others); however it does not allow us to achieve the research objectives which are oriented to find bacteria with inhibitory activity against marine pathogens which could be affecting the viability of the early developmental stages of O. mimus.

Provide sample collection time and who identified the Octopus mimus.

This species is a controlled catch species, and can only be catched in Chile between the months of April-May (Autumn) and August-October (Winter-Spring seasons in the Southern hemisphere). The samples were obtained during August in 2010. 

The samples were identified by Martha Hengst, using the diagnostic morphological characters available in the specialized literature. Particularly, chromatophores patterns in paralarvae, is one of the most valuable diagnostic characters for identification as was described in the original description of this species and later by other authors such as by Cortez 1995; Guerra et al., 1999 (Redescription of the species based on a Neotipo). It is important to note that Octopus mimus is the only shallow-water octopus species validly described for the Antofagasta Region (Chile) in the Southeast Pacific, from where the samples for this research were obtained.

We have included a photo of the paralarvae analyzed in this study in Figure 2. If the reviewers consider that we should include a big photograph of paralarvae we can include a new Figure in supplementary material.

There are several articles discussing the complexity in taxonomy and diagnostic characters of Octopus species, using classical taxonomy and molecular approaches to systematic analyses of Octopus species. We listed below some of the articles useful for the identification of O. mimus, previously not mentioned. 

Tito Cortez, Angel F González, Angel Guerra. 1999. Growth of Octopus mimus (Cephalopoda, Octopodidae) in wild populations. 1999. Fisheries Research Volume 42, Issues 1–2, Pages 31-39. ttps://doi.org/10.1016/S0165-7836(99)00040-5

Angel Guerra, Tito Cortéz, Francisco Rocha. 1999. Redescription of the Changos’ octopus, Octopus mimus Gpould, 1852, from coastal waters oh Chile and Perú (Mollusca, Cephalopoda). Iberus, 17(2): 37-57.

Warnke, K., Soller, R., Blohm, D., & U., Saint-Paul. 2000. Rapid differentiation between Octopus vulgaris Cuvier (1797) and Octopus mimus Gould (1852), using randomly amplified polymorphic DNA. J. Zool. Syst. Evol. Research, 38 (119-122.

Marcos Pérez-Lozada, Angel Guerra, Andrés SanJuan. 2002. Allozyme divergence supporting the taxonomic separation of Octopus mimus and Octopus maya from Octopus vulgaris (Cephalopoda: Octopoda). Bulletin of Marine Science, 71(2): 653-664.

Cardoso F., Villegas P. y Estrella C. 2004. Observaciones sobre la biología de Octopus mimus (Cephalopoda: Octopoda) en la costa peruana . Rev. Peru Biol., 11: 45-50.

Magallón-Gayón, E., del Río-Portilla, M.A., Barriga-Sosa, I. 2020. The complete mitochondrial genomes of two octopods of the eastern Pacific Ocean: Octopus mimus and ’Octopus fitchi’ (Cephalopoda: Octopodidae) and their phylogenetic position within Octopoda. Molecular Biology Reports, 47: 943-952.

Lines 122-123: Salinity remained stable at 35 g/L throughout the year? How did you know that? How many times did you measure?

The salinity measurement has been specified in PSU in the manuscript, and references supporting this assertion, Blanco et al. (2001) and Escribano et al. (2004), which present over 35 years of salinity monitoring data from the coasts of Antofagasta, have been incorporated in line 121.

Lines 132-133: “Mucus samples were obtained using sterile cotton swabs, which were immediately submerged in marine saline solution (Oxoid) and streaked onto Zobell agar plates” should move to Section 3.2. Isolation of microorganisms

This section was moved to Section 3.2

Lines 136-138: “Seriel dillution...” should move to Section 3.2. Isolation of Microorganisms

This section was moved to Section 3.2

Line 174: Antimicrobial should be changed to Antibacterial

This word was changed in the revised manuscript.

Provide the strain code of pathogenic strains Vibrio, and also their sources, collection, etc.

A paragraph was included in the Methods section.

Why did authors not use microdilution for antibacterial assay? Agar-based assays may not be as exact as microdilution assays.

We thank the reviewer for this suggestion. We chose to evaluate antibacterial activity in agar-based methods given that the aim of this study was to determine the presence or absence of antibacterial properties by a given isolate. Indeed, microdilution assays would provide similar information, including a range of concentrations of the bioactive compounds, but given that the nature of the compounds as well as the complexity of the array of secondary metabolites that could be secreted by this bacteria it lies beyond the scope of the present report. 

Also, it will be better if authors use ethyl acetate extract for both antibacterial assay and UHPLC analysis.

Using the extract and bacterial cells are two appropriate experimental strategies, with different levels of resolution, but for a screening antibacterial using Dopazo Method, is a low cost approach. Thanks to the reviewer for the suggestion. 

Provide more information about UHPLC analysis. The authors only mentioned how to run the equipment but did not mention how to identify chemical compounds.

A paragraph was included in the Methods Section.

I suggest that the bacterial strains should only be identified at the genus level. It is not reliable to identify them as species level only based on one gene marker.

In this stage of this investigation this is the best tool that we have to identify the bacterial strains using BLAST against this particular gene sequence, we acknowledge the limitations in this approach and further studies will aim in sequencing the genome and carry out a more detailed identification using several markers.

Provide the main figures of inhibition halo in antibacterial assays.

This was included in Figure 4 in the revised manuscript.

Why did the authors select the strain M8.1 for secondary metabolite analysis by UHPLC? Antibacterial activity of other strains such as E2.2 and E6 is better than M8.1.

Bacillus pumilus was not considered as a main species for analyses as it is considered a pathogen in the fishing industry (examples: Saggese et al., 2018; Hill et al., 2009). Also, we isolated Bacillus megaterium from eggs, which is not considered a pathogen of marine organisms. Most marine organisms have a high mortality rate during eclosion, thus it is relevant to determine the presence of bacteria with potential to inhibit the proliferation of pathogens in eggs and paralarvae, the critical developmental stages for this species. In addition, several strains of B. megaterium show biological activity as antivirals, antifungals, antibacterials, among many others, as listed in references below, most of them used as biocontrol in agroindustry; therefore would be a interesting source of metabolites in marine environments, which as been less explored.

Liu, JM., Liang, YT., Wang, SS. et al. Antimicrobial activity and comparative metabolomic analysis of Priestia megaterium strains derived from potato and d

---

## [Editor Report · Decision Letter 1]

17 Oct 2024

Epibiotic bacterial community composition varies during different developmental stages of Octopus mimus: study of cultivable representatives and their secondary metabolite production

PONE-D-24-34526R1

Dear Dr. HENGST,

We’re pleased to inform you that your manuscript has been judged scientifically suitable for publication and will be formally accepted for publication once it meets all outstanding technical requirements.

Kind regards,

Estibaliz Sansinenea

Academic Editor

PLOS ONE

Additional Editor Comments (optional):The authors have done all changes suggested by both reviewers answering their comments. Therefore the Ms can be accepted in the current form.

The authors have done all changes suggested by both reviewers answering their comments. Therefore the Ms can be accepted in the current form.
---

## [Editor Report · Acceptance letter]

13 Dec 2024

PONE-D-24-34526R1 

PLOS ONE

Dear Dr. Hengst, 

I'm pleased to inform you that your manuscript has been deemed suitable for publication in PLOS ONE. Congratulations! Your manuscript is now being handed over to our production team.

Kind regards, 

on behalf of

Dr. Estibaliz Sansinenea 

Academic Editor

PLOS ONE